# A Study on the Funerals of the Han Buddhist Monks of Lingnan during the Late Qing Dynasty via the Haichuang Temple in Guangzhou

Ronghuang Wang * and Wanqing Chen

College of Liberal Arts, Jinan University, Guangzhou 510632, China; wanching1998@stu2021.jnu.edu.cn
* Correspondence: rhwang@jnu.edu.cn

**Abstract:** The funeral protocol of Buddhist monks is an important part of the rituals of Han Buddhism. The monks' funeral rituals were recorded in detail in the Monastic Rules (清规) of Chan. The funeral of Chinese Buddhism monks after the Song Dynasty was known through the records of Monastic Rules. However, how it is concretely practiced is unknown. In the late Qing Dynasty, Westerners who came to China out of curiosity about the rituals of Han Buddhism recorded the process and details of the funerals of the monks in the temples they visited, among which Haichuang Temple (海幢寺) in Guangzhou ranks first. The funerals of the monks at Haichuang Temple in the late Qing Dynasty inherited the tradition of Chan funeral culture from the Song Dynasty. Meanwhile, the degradation into secular funeral culture appeared. Influenced by the secular funeral culture in Lingnan (岭南), the tombs of the monks in Chan Temples there, among them, Haichuang Temple is listed as a typical example, showed a trend toward the Shanshou Tomb (山手墓) in the early Qing Dynasty. In the late Qing Dynasty, some of the ancestral tomb-pagodas (祖师墓) in Lingnan Chan Temples abandoned the traditional form of pagodas completely and were almost the same as the Shanshou Tombs. The degradation of the funeral culture of Han Buddhism in the late Qing Dynasty reflects the declining trend of Buddhism.

**Keywords:** Han Buddhism; funeral rites; Haichuang Temple in Guangzhou; Buddhism in the late Qing Dynasty; monks' tomb-pagoda

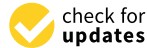



## 1. Introduction

Buddhism attaches great importance to the funeral activities of monks. There are four kinds of burial methods for Indian Buddhist monks: water burial, cremation, inhumation, and burial in a forest (林葬), which means putting the corpse in the wilderness and letting the beast prey. When Buddhism was introduced to China, "Among the four burial methods of Indian Buddhism, only burial in a forest and inhumation are heard, while cremation and water burial are rare" (东夏所传，惟闻林、土；水、火两设，世罕其踪。) (Daoxuan 2014, p. 1168). After the 9th century AD, cremation was common (Huang 1985, pp. 695–99)[1]. The Monastic Rules of Chan (清规) in the Song and Yuan Dynasties, represented by *The Regulations in Chan Monastery* (《禅苑清规》) and *Monastic Rules of the Monk Baizhang* (《敕修百丈清规》), stipulated that Chan monks should be cremated after death, and there was a set of funeral etiquette close to the secular etiquette, which has been noticed by scholars.[2] *Explanation of Monastic Rules of the Monk Baizhang* (《百丈清规证义记》), written in the reign of Emperor Daoguang, was a collection of the major achievements of the Chan monastic rules of the Qing Dynasty, which suggested that the funeral rites of abbots in the Qing Dynasty tended to degenerate (Wang 2018, pp. 96–106; Qin 2020, pp. 144–51). However, the funeral of Chinese Buddhist monks after the Song Dynasty was known through the records of Monastic Rules. However, how it was concretely practiced was unknown.

The Haichuang Temple in Guangzhou was rebuilt on the site of the Qianqiu Temple (千秋寺), which was built during the South Han Dynasty and deserted later. When rebuilt in the early Qing Dynasty, it soon became a famous Chan temple in the Lingnan area. During the reign of Emperor Qianlong, Guangzhou became the only port for foreign trade. The Haichuang Temple was appointed by the Qing government as a reception place for Western diplomatic missions to China, and it also became one of the designated scenic spots for the foreign merchants of the Thirteen Hongs (十三行) in Canton. Because of its important position in the communication between China and foreign countries, Haichuang Temple became an important subject for the export paintings of Guangzhou in the Qing Dynasty. After the Opium War, more Westerners came to China than before, and Haichuang Temple almost became a must for them. Out of curiosity about the system and rituals of Han Buddhism, the Westerners who came to China during the late Qing Dynasty recorded the process and details of the funeral of the monks in Haichuang Temple in detail. Centered on the Haichuang Temple, and taking account of the Westerners' historical records of the funerals of monks of Han Buddhism, historical documents, such as *Explanation of Monastic Rules of the Monk Baizhang*, and the related archaeological materials of tombs, this article studies the funerals of Chan monks and the shapes of the tomb-pagodas of Lingnan in the late Qing Dynasty. And the article plans to reflect on the development of Chinese Buddhism in the late Qing Dynasty by analyzing the secularization of the funerals of Chan monks in the Late Qing Dynasty.

## 2. The Funeral Rituals of the Monks of Haichuang Temple in Guangzhou in the Late Qing Dynasty in the Eyes of the Westerners in China

Due to different religious beliefs, Haichuang Temple aroused lots of interest in Westerners in China. The Westerners who had visited Haichuang Temple described its architecture, Buddhist statues, and rituals in detail in their travel notes. And more foreigners focused on the funeral rites of monks. John Henry Gray, an Englishman who came to China as a missionary in 1868, was a pastor of the Shamian (沙面) Catholic Church in Guangzhou. In 1875, he published *Walks in the City of Canton*, which recorded the custom of Guangzhou (Gray 2019, p. 3)[3]. Herbert Allen Giles, a British consular officer, appointed to China in 1867, was the first to travel to Shantou and then to Canton in 1878. In 1882, he published *Historic China and Other Sketches* to record his travels in China (Giles 2007, pp. 1–2). The Scottish traveler Constance Frederica Gordon–Cumming, who began her circumnavigation of the world in 1868, visited Guangzhou in early 1879 and described her trip in China in her book published in 1886, *Wanderings in China* (Laracy 2013, pp. 69–92). All of them visited Guangzhou in the 1870s, and their travelogs recorded the funeral rites of the monks at Haichuang Temple at that time. Although they did not witness the same funeral, their descriptions of the process and details were almost the same. The funeral ceremony of the abbot in the temple was slightly different from ordinary monks. During the late Qing Dynasty, Westerners who came to China mainly recorded the funeral ceremony of ordinary monks.

According to these three travel books and the records of other Westerners in China, the funeral activities for ordinary monks in Haichuang Temple in the late Qing Dynasty could be summarized by the following procedures: moving the critically ill monks to the Xigui Hall (西归堂), putting the body into the niche (龛), relocation of niche, sealing the niche, cremation, and putting ash into the pagoda. First of all, before the death of Chan monks, they would be arranged to live in Xigui Hall. John Henry Gray recorded that "We visited the Sai-Kwai-Tong (西归堂), or hall to which monks, when sick unto death, are taken to die. Old and infirm monks who, in consequence of their advanced age, may die at any moment, are, also, lodged in this asylum. It is a small open court yard, one side of which consists of arow of dark, damp cells. It is in these gloomy rooms, that the dying friars draw their last breath. The reason why all incurables arc, as a rule, removed to this place to die, is owing to the fact that the soul in its flight through space, to the Sai-Tien, or

western paradise, has not to traverse so great a distance, as it would have to do, were it, elsewhere, to quit its tonement of clay" (Gray 2018, pp. 81–82).

Xigui Hall, also known as Nirvana Hall (涅槃堂), Dying Place (无常院), Longevity Hall (延寿堂), Xingxing Hall (省行堂), and Sickness Hall (病堂), was located in the northwest corner of the temple for the placement of seriously ill monks. In the Nirvana Hall, there was a statue of Amitabha Buddha containing the idea of guiding critically ill monks to western paradise. According to John Henry Gray's observation, Xigui Hall of the Haichuang Temple was a very small open courtyard. The rooms inside were dark and damp.

According to *Explanation of Monastic Rules of the Monk Baizhang*, after the critically ill monk entered Nirvana Hall, he had to reflect on his own faults in order to be cured of his illness. He had to "deal with the funeral affairs, let go of all the karma, and wholeheartedly pray to Buddha" (付托后事放下万缘，一心念佛), hoping to be born in the Pure Land of Amitabha. In Nirvana Hall, there are some monks looking after them, "The patients' clothes should be washed and dried in the sun at any time, the medicinal bait should be carefully prepared, and should not be negligent or lazy" (病人衣裳宜随时洗晒，药饵宜留心煎制，不可疏忽怠惰) (Yirun 2004, p. 454). Other monks in the temple also often visit the critically ill monks to express their concern.

After the sick monk died, other monks would wash his body and put it into a niche. A "niche" was a container for the body of a dead monk. *An Encyclopedic Dictionary of Buddhism* (《释氏要览》) said "This Buddha's whole body was shaped like a pagoda, so it was known as a niche" (今释氏之周身，其形如塔，故名龛。) (Daocheng 1925, p. 307). John Henry Gray said that the "niche" resembled a wooden sedan chair which was enclosed on all sides. Entering the niche was equivalent to putting the body into a coffin at a secular funeral. John Henry Gray pointed out that "The attitude or position of the corpse, when occupying the sedan chair, is similar to that in which the idol of the past Buddha is usually represented. That is, the legs are gathered up, and crossed, the one over the other, at the base of the body, with the soles of the feet pointing towards the heavens" (Gray 2018, p. 82). The body of the dead monk should be seated in a chair, held in that position, and placed in a niche. Herbert Allen Giles observed the body of the dead monk, stating "We beheld an old man sitting bolt upright and dressed in the usual priestly garb, his hands folded before him in prayer, and his head thrown slightly back, as if he had fallen asleep". In order to keep the body in a kneeling position, "Before him, fixed in the framework of the chair itself, was a short upright piece of wood with a crescent- shaped top, intended to serve as a rest for the chin in case his head should fall forward" (Giles 1882, p. 288).

After putting the body into the niche, it was moved to the Preaching Hall (法堂). John Henry Gray noted "The sedan chair, containing the corpse, is, then, remove from the Sai-Kwai-Tong, through the doorway-the gate of death- and conveyed to a neighboring fane, and there placed upon an altar" (Gray 2018, p. 82). Then, the monks set up a memorial tablet (灵位) for the dead monk in Preaching Hall. According to Herbert Allen Giles, "A narrow strip of yellow paper, bearing certain characters upon its face, was pasted on the slide of the box, and that a table was arranged in front with several plates of food, etc., and a taper burning at the side". He said, "Above the box and altar were two Chinese characters said that he was on his return journey to the west, to the land of Buddha; in other words, he was dead. There was a scroll hanging on each side, on which were the following words: 'Though the Trikâya be absolutely complete, the limit is not yet found'. 'It is the maturity of the Skandha+ which alone can give perfection'. The yellow strip of paper pasted on to the vertical slide above mentioned bore this inscription: 'The throne of intelligence' of the contemplative philosopher, the Bôdhisatva, the worthy Bikshu 'United Wisdom,' now passed away" (Giles 1882, p. 288).[4]

Then, sealing the niche began. The ritual holder recited the gatha and sealed the niche.[5] The content of the gatha generally contained the activities, praises, and regrets of and for the deceased. Once sealed, no one was allowed to reopen the niche. Herbert Allen Giles and his companions had wanted to look carefully at the remain but were refused.

The niche would generally stay in the Preaching Hall for twelve hours during which time the monks intone sutras (Gray 2013, p. 293). After twelve hours, the preparation for the cremation began. Bringing the niche to the furnace for the cremation usually began after breakfast. Before lifting the niche, the monks would chant The Amitabha Sutra and sing praises for the dead. Herbert Allen Giles recorded, "While thus engaged we heard the harsh tones of the 'wooden fish,' beaten to summon the priests to their morning meal, and about a quarter of an hour afterwards they begun one by one to drop in, each with his kachaya or coloured stole hanging in readiness over one arm. Then ensued a series of prostrations on the circular rush mat placed in front of the altar and coffin. Just at that moment it was announced that the abbot was coming; and immediately all the priests put on their Stoles, and arranged themselves decorously in two long rows, beginning from close alongside the coffin itself. In a few minutes the abbot was passing slowly between their ranks, his string of 108 beads in one hand, and in the other a small gong fixed into a framework of wood, having a clapper so attached that every turn of the hand produces a sound. He stopped in front of the altar and coffin, and there prostrated himself thrice, each time knocking his head upon the ground thrice, that being the number of obeisances performed before the Emperor of China, in the presence of death, and on other special occasions" (Giles 1882, p. 289). After that, they began to go to the cremation furnace.

According to the description of Herbert Allan Giles, the funeral procession led by two kids was not short. "Each was confided a streaming banner attached to the top of a light rod, ornamented with a blue and white spiral from top to bottom. Both banners bore the same legend: 'Our humble trust is in Amida Buddha, our guide' (Giles 1882, p. 290). The leading kids were actually the little monks and Sanskrit. The banner they were holding was a streamer for the funeral procession, which was also called a "four-fundamental streamer (四本幡)" (Ciyi 2005, p. 5981). Behind them was the niche followed by the funeral host and other monks. According to Constance Frederica Gordon–Cumming, the monks joining the procession usually wore sackcloth, with a white cloth wrapped around their heads (Constance Frederica Gordon-Cumming 1886, p. 90).

The troop arrived at the furnace for the cremation, and John Henry Gray described the process and the furnace below.

> The funeral pyre upon which the mortal remains of priests are burned, occupies one corner of this garden. It is built of bricks, and, in form, resembles a small domed tower. It is approached by an open doorway, which, in point of width, is sufficiently large to admit the wooden sedan chair in which the corpse, awaiting cremation, is contained. The sedan chair, with the precious dust, which it contains, is, when taken into the tower to which, as the funeral pyre, we have just referred, placed on four stones, and around it faggots, in large quantities, are immediately piled. The priests, who form the funeral procession, then arrange themselves in front of the pyre, and, for the repose of the departed soul, proceed to chant a requiem. On bringing to a close the first portion of this religious ceremony, the senior priest of the funeral party, having received at the hands of a secular brother a lighted torch, applies it to the faggots, which, for the cremation of the corpse, have been set in order. As the flames burst forth, the monks again engage in religious services, which are continued until the remains of the departed one have been consumed. The ashes, so soon as they have become cold, are gathered together, and deposited in a cinerary urn. (Gray 2018, pp. 79–80)

The wooden sedan chair written by John Henry Gray was the "niche". Before cremated, the monk who handled the funeral would chant some gatha that was easy to understand to express the wish for the deceased. And he would also use the torch to knock on the niche in this way to remind the deceased of his entrustment.

The last step was putting the ash into the pagoda. Initially, the ash would be put in the neighboring pagoda courtyard (塔院) of the Common Pagoda (普同塔). According to the notes of Martha Noyes Williams, who came to Guangzhou in the 1860s, the cremains would stay there for some time and, after that, would be allowed to be taken to be buried

outside by the deceased's kinsfolk. "If not taken by relatives, after a reasonable length of time, to be buried among their kindred, are finally interred in the cemetery" (Williams 1864, p. 205). The so-called "cemetery" is the Common Pagoda. Just as John Henry Gray and Constance Frederica Gordon–Cumming pointed out, in special festivals, such as the Qingming Festival or Double-ninth Festival, the monks in Haichuang Temple would put the ash into bags made of red cloth and place them in the Common Pagoda within one year.

### 3. The Characteristics of Chan Monks' Burial Rituals in the Late Qing Dynasty Reflected by Funeral Rites of the Monks in Haichuang Temple

*Explanation of Monastic Rules of the Monk Baizhang* written by Yirun (仪润) is the literature reflecting the Buddhist systems and Monastic Rules of Chan in Han Buddhism of the Qing Dynasty. In chapter 5, the whole process of burial rituals of abbots and ordinary monks in temples is recorded. This chapter will compare the funeral practices in Haichuang Temple in the Late Qing Dynasty with the practices recorded in the book to reveal the generality and particularity of funeral practices in Haichuang Temple in the Late Qing Dynasty.

According to the stipulations of *Explanation of Monastic Rules of the Monk Baizhang*, when the abbot was suffering from a critical illness, he would move to Eastern Hall (东堂). "When the moment came, anyone who accompanied the abbot would unanimously recite the Buddha's name in union so as to help him be born in the Pure Land of Amitabha" (临寂时至，凡伴病者齐声念佛，以助往生。). If the abbot entered the stillness of Nirvana (died), his funeral would begin after some time (equated with an incense burnt cost一炷香的时间). And the funeral includes dressing for the dead after washing his body, and then putting it into a coffin. The niche was kept in a room of the abbot, and a shrine was set up where people can pay homage. The shrine was guarded by the little disciple who had been tonsured by the abbot. When the abbot was dead, the sealing niche was performed by the abbot from another temple. "The niche was kept 21 days" (停棺三七日), during which time monks were asked to recite *The Amitabha-sutra.* After the breakfast on the funeral day, the funeral started, which involved lifting the niche, the procession walking over to the furnace (the "Platform of Nirvana", i.e., a furnace for cremation), and the abbot from other temples hosted the torch raising and collecting bone ashes, etc. The bone ashes of the abbot would be put into the pagoda the next day. "The pagoda is as high as about 40 inches, not extravagantly built, no stonemason and other meaningless things, such as color pavilion, music, etc., will not be used, but only to recite Buddha's name by mass monks". (塔高三尺，不得侈费，石工及举殡无益之事，如彩亭、音乐等，俱不应用，但众僧念佛导引而已). During the funeral the disciples "only feels deeply sad" (但心丧而已) and did not follow the secular trend of wearing mourning and crying in a funeral. If the dead monk was someone in charge of temple affairs (两序执事), when critically ill, he would move to the Guest Hall (客堂). After death, his niche was also put in Guest Hall; when ordinary monks were critically ill, they would be allocated to Xingxing Hall, after death putting their niche in there but without offerings. When lifting or sealing the niche, the abbot of this temple spoke of gatha, and when cremation began, the abbot would hold the torch. The day after cremation, the Common Pagoda was opened, and the ash wrapped in a container by a rope would be lowered down into it, and then it was closed.(Yirun 2004, pp. 339–45).

The funeral ceremony of Buddhist monks in Haichuang Temple in the late Qing Dynasty is basically consistent with the process written in *The Explanation of Monastic Rules of Monk Baizhang*. However, the follow-up parts after cremation were missing, which included the evaluation and auction of the relics of dead monks (估唱) and sending the spirit tablet to the Ancestral Hall (祖堂) (ordinary dead monks would be just added their names to the public tablets), which may be overlooked by the Westerners in China. The funeral practice corresponds with the regulations speculated by *Explanation of Monastic Rules of the Monk Baizhang*, reflecting a succession of Chan funeral culture. The literature of the Song and Yuan Dynasties reflected the general funeral practice for ordinary monks. When

the monk was dead, the niche containing his body was placed in Longevity Hall, and the monks recited the sutra, locked the niche, and the next morning carried the niche, held the torch, collected bone ashes in Longevity Hall, then waited for the right time to carry them to the Common Pagoda. As for the abbot, the niche containing his body was placed in the abbot's room, and then the niche was allocated to Preaching Hall and sealed, and in the hall, the portrait was hung for mourning, paying respect, and offering tea (奠茶汤,offering tea in front of the memorial tablet). Then the host faced the spirit tablet (灵位) and a special sutra was recited, which was named "Xiaocan" (小参), and tea was offered. The funeral day came, the niche was carried, and the portrait was hung on temple gates and tea was offered. If the body of the dead abbot was cremated, the respectable elderly monk would hold the torch, then the bone ashes would be collected in the bedroom and hung at the portrait to enshrine (挂真供养), waiting for the right time to carry them to the Pagoda. If the body of the dead abbot was carried to the Pagoda, the respectable elderly monk would place the niche down into the earth and scatter the sand, and then the portrait was welcomed to the bedroom to enshrine (Zongze 2006, pp. 85–87, 95–96; Weimian 1293, pp. 611–14; Yixian n.d., pp. 63, 652–58; Dehui 2006, pp. 79–92, 171–78). The funeral procedures of the Chan in the Qing Dynasties canceled the offering and tea, the reciting of "Xiaocan" to the spirit tablet, and the hanging of the portrait on the temple gate, and others did the same as the Song and the Yuan Dynasties basically.

Some details of the funeral ceremony at Haichuang Temple in the late Qing Dynasty also followed the Chan customs since the Song and Yuan Dynasties. For example, the day for the sealing of the niche came three days after the abbot was dead, and the day chosen for the funeral was up to "the amount of property, the weather, for ten to half a month, all depending on whether the things go smoothly" (Yixian n.d., p. 654). The ordinary monks applied the practice of the funeral on the day after their death. *Explanation of Monastic Rules of the Monk Baizhang* stipulated that when the abbot was dead, "the niche should be kept for 21 days"; when the affairs monk was dead, his niche was relocated and kept there for four days, and then a funeral began; for ordinary monks, the funeral began the next day after their death (Yirun 2004, p. 341). For ordinary monks in Haichuang Temple, the funeral began after breakfast the next day after their death, which was also a tradition passed down from the Song and the Yuan Dynasties.

Of course, there are also some details that differ from the record in *Explanation of Monastic Rules of the Monk Baizhang*. The book emphasized that when the monk was dead, "the disciples whom he tonsured must not wear mourning and wail", and were "not allowed to decorate mourning hall, put up elegiac couplet, wear mourning, issue obituary everywhere, assemble believers and relatives, and degrade to secular funerals". ("遗戒小师不得披麻恸哭"，"不张罗孝堂，不广列联挽，不披麻戴白，不四出报讣，不纠集施主眷属，不作俗格道场") (Yirun 2004, pp. 339, 341).In actual practice, individual temples differed. At the funeral of Haichuang Temple, the Preaching Hall had banners, elegiac couplets, and offerings, and the procession wore mourning clothes in a very secular way. While at Yongquan Temple in the Gu Mountain (鼓山涌泉寺) of Fuzhou in the Late Qing Dynasty, the monks were not seen to wear mourning clothes, but a "light purple robe, and yellow kasaya put on shoulders" (Lu 2018, pp. 260–61). After the Tang Dynasty, the funeral for monks resembled secular ones in many aspects. Relevant monks chose mourning attire appropriate to ritual systems and paid homage to the deceased in front of the catafalque. If the master died, his disciples would behave in the same way as a son to his dead father at a funeral. Even though these practices suffered some criticism, the phenomenon that the Chan monks observed mourning for their masters in the Qing Dynasty was still quite common.

The book *Wu Shan Lian Ruo Sin Shue Bei Yong* (《五杉练若新学备用》) completed in Five Dynasties stipulated that in order to differ from the secular people, the body of a dead monk could be put in the niche in a position of either sitting or lying. "If he dies lying, he lies on the matting with his breast, head in the north and face to the west, with quilts covered on his body; if he dies sitting, the shengchuang (绳床) is put facing south, the

central hall is put up by incenses, light, and tea, as waiting for the body put into the niche" (卧终，即右胁著席，北首面西，以衾覆之；若座（坐）亡，即绳床面南，中堂设香灯茶，以候入龛柩。) (Yingzhi 2018, p. 563). Many monks in the Qing Dynasty sought the fame of "die sitting". They arranged dead monks died sitting and put the bodies into the niche in the sitting position. *Explanation of Monastic Rules of the Monk Baizhang* believed that monks who chose to die sitting were contrary to ritual laws, so it criticized that "recent 'die sitting' being too prevalent" (近日坐龛一事，相习成风), and advocated obeying the old regulations by letting the monks die lying and putting the body into the niche in the lying position (Yirun 2004, p. 341). Although this had been said, Haichuang Temple did not obey the book, it followed the traditional sitting position way, which was the same case as in Yongquan Temple in Gu Mountain.

*Explanation of Monastic Rules of the Monk Baizhang* argued that cremation was a Buddhist system good for Dharmakaya (法身), criticizing "Now there are even some monks afraid of cremation and leaving wills not to do so. How stupid it is! There goes the fake news that abbot need not being cremated after death. Both of them are totally wrong!" (今竟有僧畏烧化而遗命不烧者，愚之甚矣。又误传为一代住持者，则不烧化，亦讹。) (Yirun 2004, p. 342). Contrary to what was required by *Explanation of Monastic Rules of the Monk Baizhang*, the monks in Haichuang Temple could decide whether they accepted the cremation or not. Alice M. Frere visiting the temple in the 1860s, pointed out that "Burning of the bodies is not a sine quâ non, as those who prefer it are allowed to be buried, on expressing their wishes before death" (Frere 1870, pp. 229–300). That was to say burial in the ground without cremation had a long history. Constance Frederica Gordon–Cumming, who visited Haichuang Temple in the 1970s, pointed out "Ordinary burial in ponderous coffins is lawful even for a priest. Such cases, though rare, have occurred in comparatively recent years, and some very old horse-shoe tombs in the temple grounds prove that such burials were permitted long ago" (Constance Frederica Gordon-Cumming 1886, p. 90). A British missionary, Robert Morrison, who came to China during the Reign of Emperor Jiaqing, said that not all the dead monks in Haichuang Temple were cremated that way; the monks who had a lot of property could build individual tombs without cremation (Morrison 2011, p. 984). What is worth mentioning is that the burial in the underground of Haichuang Temple originated from the Chan tradition, which had the burial convention of "putting the whole body in the pagoda" (全身入塔). The body was not being cremated, but digging the ground where the body was buried into a cellar with stones served as floor, and then the niche covered it with stones and finally buried it all with earth, and then erected a stone pagoda there (Daozhong 2004, p. 627).

According to *Explanation of Monastic Rules of the Monk Baizhang*, the temple courtyard had different systems, which can be three pagodas, five pagodas, seven pagodas, and nine pagodas. The system of three was a pagoda of abbot (祖师塔) on the central, Bhiksu from this and other temples, and the Common Pagoda for Sanskrit from this and other temples on the right. "The Pagoda for Buddhist nuns still belongs to nunnery, not allowed to affiliate itself to temples for monks" (其尼塔仍归尼庵，不许附僧寺也), it criticized that "recently the Pagoda for nuns mostly affiliated itself to temples for monks, every time the tomb sweeping crosses, and the drawbacks easily seen" (迩来尼塔大都附于男寺，不时祭扫往来，流弊百出) (Yirun 2004, p. 344). As Alice M. Frere pointed out, on one side of the pagoda courtyard of Haichuang Temple was a Common Pagoda for home monks, and on the other side was a Common Pagoda for nuns in the neighboring nunnery (Frere 1870, pp. 229–31). This was exactly what *Explanation of Monastic Rules of the Monk Baizhang* criticized.

As deduced from above, in terms of procedures, the funeral rituals for monks in Haichuang Temple in the Late Qing Dynasty corresponded with *Explanation of Monastic Rules of the Monk Baizhang*, some of which originated from the funeral tradition of the Song and Yuan Dynasties. Meanwhile, Haichuang Temple still used some practices that *Explanation of Monastic Rules of the Monk Baizhang* had criticized, which reflected not only the peculiarity of Haichuang Temple in terms of funeral culture but was characteristic of an

age in which the funeral rules and regulations of Han Buddhism of the late Qing Dynasty gradually went against what *Explanation of Monastic Rules of the Monk Baizhang* had set.

## 4. The Shapes of Tomb-Pagodas for the Monks of Lingnan in the Late Qing Dynasty

The tomb-pagoda (墓塔) is the place for burying the remains of monks (it is called tomb-pagoda because of its pagoda-shape). There are two main kinds: one is the ancestral pagoda, which is specially built for patriarchs initiating or reviving a Chan Sect, or venerable masters. They can be arranged in the form of cremation or a whole-body burial after departing the world. Another is a Common Pagoda, which is for other abbots, and abbots and common monks are buried together after they were cremated (Zhou 2013, pp. 99–103). Chan Master Yunju Yuanyou (云居元佑) in the Northern Song Dynasty had initiated the three-pagoda system. It can be divided into three: the pagoda for patriarchs and masters with measureless merit and virtue, the Common Pagoda for other abbots, and the Common pagoda for commoners. From then on, the three-pagoda system had been gradually and commonly used by the Chan School, the Pope School, and the Vinaya School, and it had become the norm for Han Buddhism in the southern Song Dynasty (Wang 2020, pp. 27–30). As the secular trend of elaborate funerals prevailed, some abbots were reluctant to be buried in the Common Pagoda for common abbots, and tomb-pagodas were even built for some of them by their disciples and followers to manifest their merit and virtue, thus the Cemetery of monk (塔林) came into being.

During the Ming and Qing Dynasties, there are several architectural styles of pagodas in China, each with their own distinctive features. The lama-style pagoda, also known as the covered-bowl-style pagoda, consists of a circular base supporting a rounded tower body, adorned with carved circular wheels on the top. The pagoda spire is adorned with an umbrella canopy and a precious jewel. This architectural style is influenced by Tibetan Buddhism and is exemplified by the Ming Dynasty pagoda of Monk Zhudong Wuwan (竺东悟万) in Shaolin Temple (少林寺). Another style is the multiple-eave-style pagoda, characterized by a pedestal and a multilayered eave structure above the tower chamber. This style is more prevalent in northern regions, and notable examples include the Ming Dynasty tomb-pagoda of Yao Guangxiao (姚广孝) at Changle Temple Village (长乐寺村) in Beijing and the Qing Dynasty pagoda of Monk Bian Haikuan (彼岸海宽) at Shaolin Temple. The scripture-tower-style pagoda, represented by the Ming Dynasty pagoda of Chan Master Wuyi (无疑禅师) in Yangqu County (阳曲县), Shanxi Province, features a base platform as the pedestal, a two-story tower body with an octagonal projecting eave, and a spire composed of three levels resembling an upturned lotus and a bottle gourd shape (Li and Cui 1999, p. 48). The pavilion-style pagoda adopts the architectural style of a pavilion and can be square with a single eave, hexagonal with multiple eaves, or octagonal with multiple eaves. Only the first story of the tower has a tower chamber, while the others do not (Zhang 2000, p. 66). Lastly, there is the oviform-style pagoda (卵塔), which follows the Chan Buddhist tradition of the Song Dynasty. Originally, it consisted of a pedestal at the bottom, an upturned lotus seat in the middle, and a tower body on top. In the late Ming Dynasty, the tower body gradually transformed into a melon ridge or square shape, covered with a precious canopy. The Vinayacharya Xingzhi (性祇律师) Pagoda in the Jiechuang Monastery (戒幢律寺) in Su Zhou, dating back to the late Ming Dynasty, and the Ten-Direction Common Pagoda (十方普同塔) in Shangfangshan Monastery (上方山塔院), from the Qing Dynasty, are exemplary instances.

The ancestral tombs group of Haichuang Temple lies in today's Butterfly Ridge (蝴蝶岭) of Jingu Hillock (金鼓岗) on Baiyun Mountain (白云山) in Guangzhou City. The fourth survey of cultural relics in Guangzhou found that it consists of 26 tombs. The tombs are built with gray sand and were renovated in the 1980s, most of which were furnished with cement. Every tomb was mainly made up of tombstone(墓碑), ridge protection (护岭), loop protection (环垄), a platform for worshiping (拜台), mountain flanks that were like two hands of a person (山手), and Houtu (后土，Tuti for tombs). Among these tombs, the bigger one also consists of drum-shaped stones (石鼓) and a moon lake (月池) (see Fig-

ure 1) (Chen 2008, pp. 120–29). This kind of tomb had the shape of Shanshou or mountain hands(山手), which was far different from the tradition of Han Buddhism in terms of the shapes of tomb-pagodas.

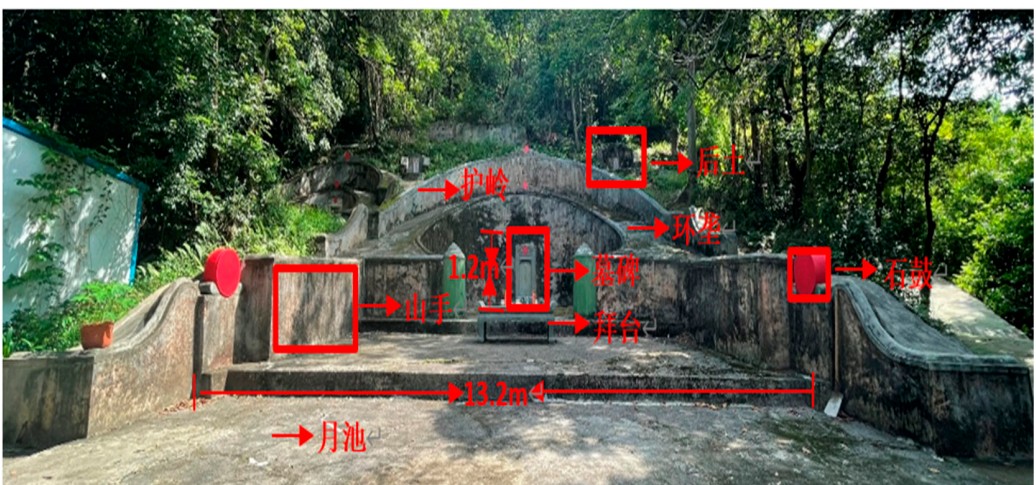

**Figure 1.** Tomb-pagoda of Guangmou (光牟) and Chiyue (池月) in the Ancestral Tomb-pagoda Group of Haichuang Temple.Photograph by the author.

The Tomb of Mountain Hands, also called Shanshou Tomb, gets its name for its semi-circle shape, resembling a person holding two hands out. It originated from the brick-built tomb of the Han Dynasty. It appeared in the Song Dynasty and started to gain popularity in the Ming Dynasty. In sharp contrast to the tomb in the northern area, which is shaped in a steamed bun hummock, the cell was built on the surface ground leaning against a ring-shaped protection ridge stretching from behind to the front in a gradually declining degree, forming a slope. Because the shape likes the chair of Taishi (太师) was also called "the Tomb of the Chair of Taishi" (太师椅墓), "the Tomb of Folding Chair" (交椅坟), or "the Tomb of Chair" (椅子坟). According to research, the Tomb of Mountain Hands was widespread in the southeast and central part of Zhejiang Province, and parts of Jiangxi, Anhui, Guangdong, Guangxi, and Fujian Province. In Taiwan and Fujian Province, there was also a kind of tomb in the shape of mountain hands, but its top resembled a tortoise shell, and it was called Guike Tomb (龟壳墓). In essence, it was a variety of the Shanshou Tomb (Zhou 1995, pp. 146–50). In Baiyun Mountain, there are many such Shanshou Tombs dating back to the Song Dynasty, but all have been repaired. Now, it is very hard to recognize its original form. In the Ming and Qing Dynasties, the Shanshou Tomb was sought-after in Guangzhou. Located on the east foot of Tianping Mountain (天平山) in the Zengcheng District (增城区), the tomb of Zhan Ruoshui (湛若水), built in the 42nd year of Emperor Jiajing's reign (1563), is an existing relatively early one.[6] There were also many secular tombs taking the shape of Shanshou. For instance, the joint burial tomb (合葬墓) of Liang Peilan (梁佩兰) and his wife, Liang was one of the "three masters in literature of Lingnan" (岭南三大家) in the Qing Dynasty and was located at the south foot of Kezi Ridge (柯子岭). It was built in the 50th year of Emperor Kangxi's reign (1711) with gray sand composed of a protection ridge, loop protection, a platform for worship, mountain hands, and moon lake (see https://zh.m.wikipedia.org/wiki/File: %E6%A2%81%E4%BD%A9%E5%85%B0%E5%A2%93.jpg, e.g., accessed on 24 May 2023) (Chen 2008, p. 202). It is not hard to see that the ancestral tomb-pagodas of Haichuang Temple were modeled after the secular tomb of the Lingnan area.

On either side of the principal cell of the Shanshou Tomb was a stone wall called "Guabang" (挂榜), some of which was engraved with the names of the buried. And the tomb of Liang has two stone tablets, each on the left and the right side. It is a pity that the characters had been peeled off and we do not know the accompanying people's information. There are a large number of Shanshou Tombs concerning the joint-burial for a couple and

even the multi-burial for a family in Baiyun Mountain. And there are 10 tomb-pagodas for multi-burial in the ancestral tomb-pagoda group of Haichuang Temple. For example, built in the 2nd year of Emperor Tongzhi's Reign (1863), the tomb of the eleventh generation of Boshan (博山, another name for monk Wuyi Yuanlai) Offspring, Chan Master Eryan Yingdi (二严应谛禅师) (see Figure 2) was built with two Guabangs on each side, which was made into the part of the loop protection. And each Guabang was engraved out of a frame of a tomb tablet made of stone. The left banner carved the words "the Pagoda of Haichuang's Tenth Generation (the quoter: the 12th generation) of Master Chonggeng" and was erected in the 3rd year of Emperor Guangxu (1877). Engraved on the right side of the tomb was "the Pagoda of Haichuang's 12th Master Juhanhai, which was erected in the 15th year of Emperor Guangxu (1889)". The companions were Yingdi's disciples, Chonggeng Dayi (崇庚达颐), and Juhan Yuanhai (巨涵源海) Companions of other ancestral tomb-pagodas revealed the disciples of the tomb-pagoda owner.[7] The multi-burial of the master and disciples in Haichuang Temple was obviously inspired by the secular funeral rites and was also out of concern for saving land resources and reducing building costs.

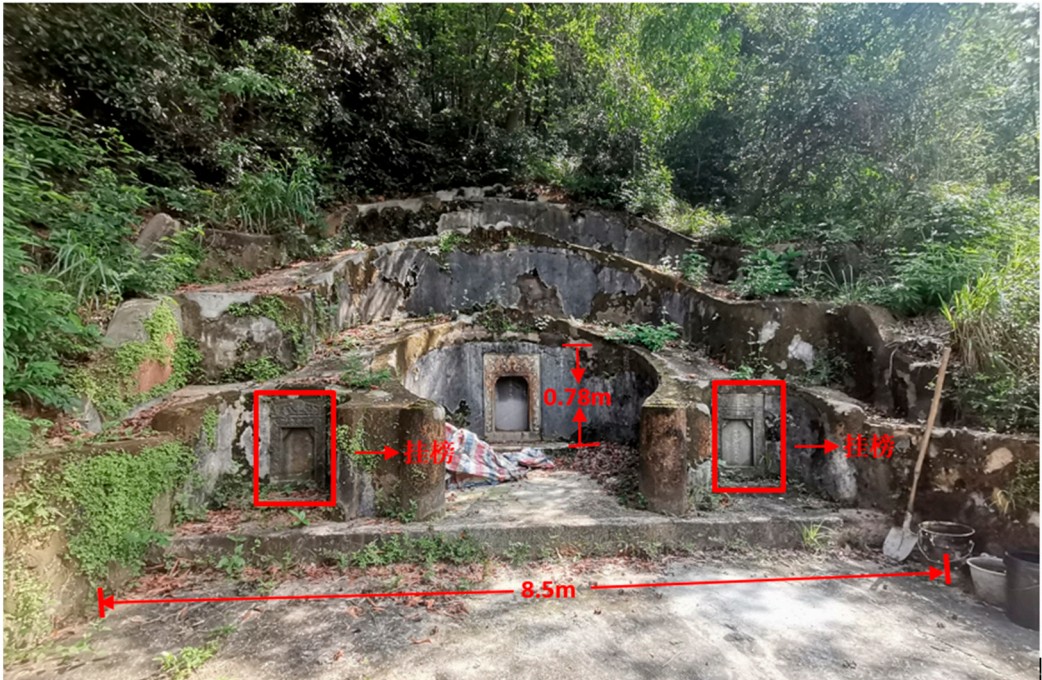

**Figure 2.** Tomb-pagoda of Eryan Yingdi in the Ancestral Tomb-pagoda Group of Haichuang Temple. Photograph by the author.

The words on the tablet of the ancestral tomb-pagoda group of Haichuang Temple had been peeled off due to long years of erosion and human destruction. So, the owner of the tomb was unknown to us. Afterward, home monks repaired tomb-pagoda and erected a new tablet, but sometimes they might be wrong. For example, a newly made tablet had written "the Tomb-pagoda of the Chan Master Jinwu (今无禅师) under Boshan Offspring of Caodong Sect", and "the 19th year of Emperor Kangxi (1680)". In fact, Jinwu died in the 12th year of Kangxi's Reign (1681). Jinwu played an important role in the development of Haichuang Temple in the Qing Dynasty. And it also could be proved wrong that it had a cramped layout that could not be used for the patriarch who had initiated and founded the temple. All the tomb-pagodas were rebuilt in 2004, the largest one's newly made tablet stated "the Ming Dynasty Ancestral Tomb-pagoda of Guang Mou and Chi Yue at Haichuang Chan Temple". Another newly made tablet stated "the Pagoda of Chan Master Jinchuan (今传禅师) under Boshan Offspring of Caodong Sect", and "auspicious year auspicious day (a fortunate day) in the Reign of Emperor Shunzhi of Qing Dynasty", from which we could see both titles of the owners of the two tablets, and the times for the

erection of the tablets were not in line with the ancient system. Hence, it remained uncertain whether the so-called "the ancestral tomb-pagoda group of Haichuang Temple" is the real place of "the ancestral tomb of Haichuang Temple in early Qing Dynasty".

In June 2023, we conducted research on the ancestral pagodas of Haichuang Temple and discovered 13 new ancestral pagodas. Therefore, the total number of ancestral pagodas in Haichuang Temple is estimated to be at least 40. Among the newly discovered pagodas, five are built during the reign of Emperor Qianlong, two during the reign of Emperor Jiaqing, one during the reign of Emperor Daoguang, one during the reign of Emperor Tongzhi, and one during the reign of Emperor Guangxu. The ancestral pagodas built after the reign of Emperor Jiaqing follow the Shanshou tomb style. The Qianlong-era ancestral pagodas are stone-standing pagodas with a pedestal in the form of a base, a square tower body with a niche on the front, a spire with a lotus-shaped pedestal, and a four-cornered pointed top. The stone pagodas are surrounded by loop protection, platforms, and mountain flanks (see Figure 3). The associated buildings of the Wuben Fayi (无本法一) Pagoda also include symmetrical stone lions (see Figure 4).

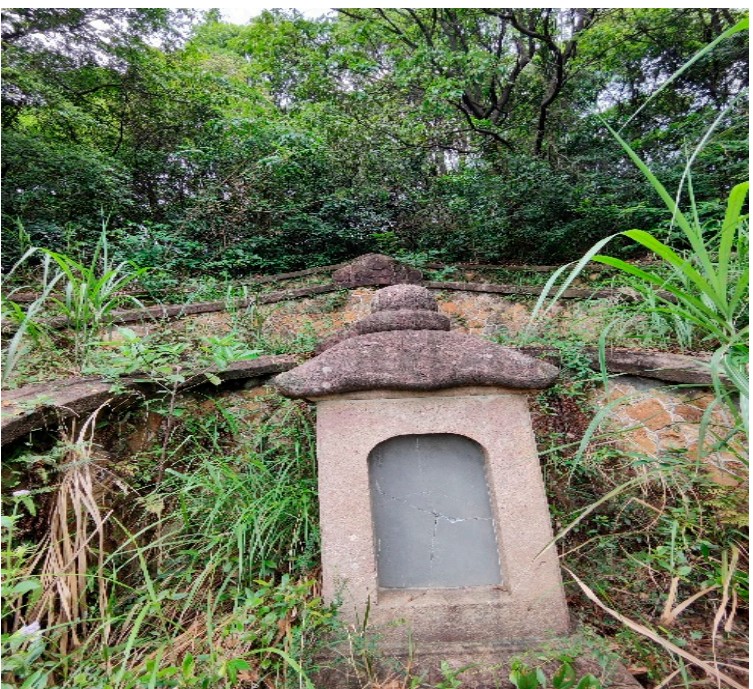

**Figure 3.** Tomb-pagoda of Xutang Xindan (旭堂心旦) in the Ancestral Tomb-pagoda Group of Haichuang Temple. Photograph by the author.

Overall, the ancestral pagodas of Haichuang Temple in the Qing Dynasty exhibit distinct chronological changes. The Qianlong-era pagodas are stone pagodas with associated Shanshou grave structures, with five remaining examples. The Jiaqing and subsequent periods adopted the Shanshou grave style. It is estimated that there are three pagodas during the reign of Emperor Daoguang, four pagodas during the reign of Emperor Xianfeng, ten pagodas during the reign of Emperor Tongzhi, and fifteen pagodas during the reign of Emperor Guangxu. The phenomenon of collective burials in pagodas emerged during the reigns of Emperor Tongzhi and Emperor Guangxu.

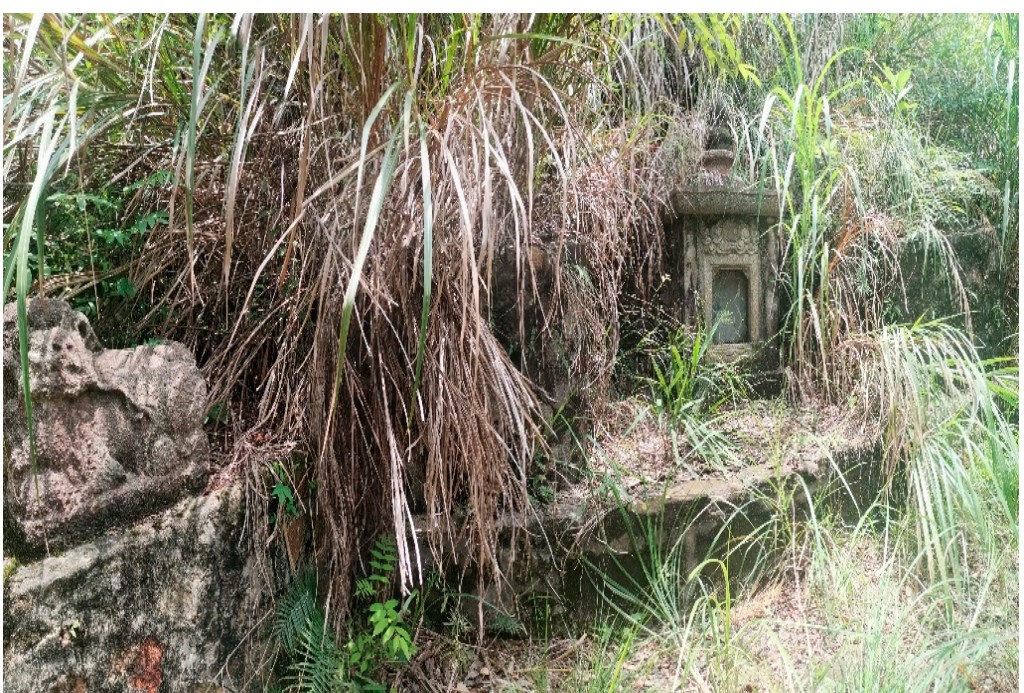

**Figure 4.** Tomb-pagoda of Wuben Fayi in the Ancestral Tomb-pagoda Group of Haichuang Temple. Photograph by the author.

What needs to be mentioned is that the ancestral tomb-pagoda taking the shape of the Shanshou Tomb is not the only one that exists in Haichuang Temple. Located in the Erlong Valley (二龙谷) of Baiyun Mountain, the ancestral tomb-pagoda group of Hualin Temple (华林寺) now has 11 tombs, and many of them have been broken (Chen 2008, pp. 114–19). According to the cultural relic survey, Tomb No. 5 is preserved in a relatively complete form. The Shanshou Tomb is built with gray sand, made up of Houtu, ridge protection, loop protection, a platform for worshiping, mountain flanks (like two hands of a person), and stone drums. The green stone tablet was engraved with the words "the Pagoda of Old Monk Zhengchijian (正持鉴), Abbot of Hualin". It was erected in the 16th year of Emperor Guangxu (1890) and took almost the same form as the ancestral tomb-pagoda of Haichuang Temple (see https://mp.weixin.qq.com/s/qoYDfsGoMP4P6QSNwvpliQ, e.g., accessed on 22 May 2023). Tomb No. 4 is the pagoda of the Abbot Lihuan Yuanjue (离幻圆觉), the second abbot of Hualin Temple and the thirty-third generation of the Linji Sect. It was erected in the 22nd year of Kangxi (1683), and the right mountain hand remains in broken parts, but the ridge protection and loop protection have been damaged. As opposed to the Shanshou Tomb, in the center of the loop protection there is a stone pagoda with an engraved tablet in the front. Tomb No. 8 has the largest layout, with ridge protection, loop protection, a platform for worshiping, mountain flanks, and a platform base. In the center of the loop protection there is a stone pagoda with an engraved tablet in the front that reads "the pagoda of the Abbot from Qingyun Temple in Dinghu Mountain (鼎湖山庆云寺) and Hualin Temple, the 34th generation of Linji sect". As for the descent, the owner of No. 8 is equal to the disciples of Lihuan Yuanjue, i.e., the time for building the No. 4 Tomb is a little later than the former, and the shape is similar to the former (see https://mp.weixin.qq.com/s/qoYDfsGoMP4P6QSNwvpliQ, e.g., accessed on 22 May 2023) (Chen 2008, pp. 114–19).

The ancestral tomb-pagoda group of Liurong Temple (六榕寺) in Guangzhou also existed as a contrasting phenomenon between the early and late Qing Dynasty in the aspect of the shape of the tomb-pagoda. Located in Heshun Hillock (和顺岗) of the Kezi Ridge in Baiyun Mountain, the tomb-pagoda group preserves eight Shanshou Tombs and eighty-five tomb-pagodas (Chen 2008, pp. 130–45). According to the cultural relic survey, Tomb No. 2 is made up of loop protection, a platform for worshiping, mountain flanks, a

moon lake, and drum-shaped stones, and was built with stones and gray sand. Resembling Tomb No. 8 in Hualin Temple, in the center of the loop protection there is an octagonal stone pagoda with a spike head and multi-layer tower (see Figure 5). A tablet is engraved in the front of the stone pagoda, stating "the Pagoda of the Abbot Old Monk Kongyin (空隐老和尚) in Qing Dynasty". On the left and right sides of the tomb, there are two stone tablets, which state "the Pagoda of Old Monk Jueshi (觉师老和尚)" and "the Pagoda of Old Monk Yuejiao (月皎老和尚)". Attached with a brief biography, both were erected in the 11th year of Emperor Yongzheng (1733). Apart from No. 1, the Pagoda for Chan Master Ze Anmian (则安棉禅师) was newly built in 2002, and six other pagodas were erected or rebuilt in Guangxu's reign, both as the typical Shanshou Tombs, which had their tablets erected in the center of their loop protection (see Figure 6). The former 85 pagodas in their originality were all destroyed. As seen from reconstruction, tomb-pagodas were rebuilt as single building without such loop protection, a platform for worshiping, or mountain hands as auxiliary architectures. The base of the tomb-pagoda is two octagonal or square layers. The body is up-wide and down-narrow-shaped in columns or squares. A tablet was embedded on the front side. The pagoda has many different shapes such as a lotus-shaped tower, four corners of the spire, octagonal spire, etc. (see Figure 7).

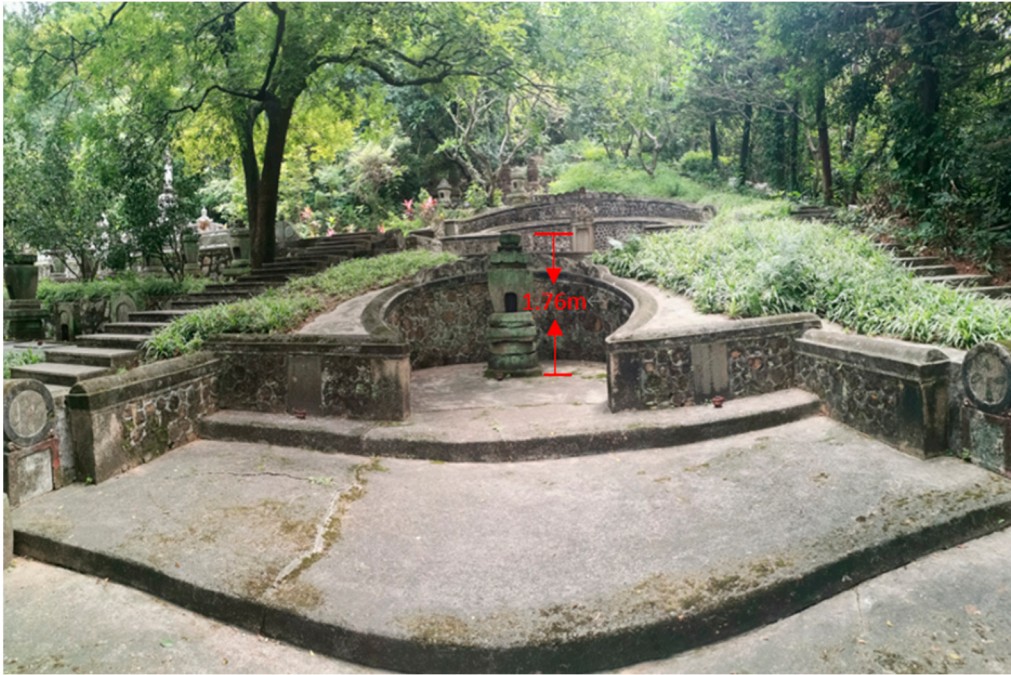

**Figure 5.** Pagoda of the Abbot Old Monk Kongyin of Liurong Temple. Photograph by the author.

The ancestral tomb-pagodas of Chan School in the early Qing Dynasty exist in other areas of Lingnan and have a similar pattern to Hualin and Liurong Temples. The examples are the first abbot Qihe Daoqiu (栖壑道丘) Pagoda in Qingyun Chan Temple in Dinghu Mountain, and the second abbot Zaisan Hongzan (在犙弘赞) (see Figures 8 and 9). They were built in the 17th year of Shunzhi (1660) and the 30th year of Kangxi (1690), respectively. Both of them were composed of loop protection, a platform for worshiping, and mountain hands in the center. Loop protection erected a stone pagoda with a Sumeru pedestal (须弥座) as the base, and the body was square. There was an engraved tablet in the front and the top was a cover shaped like a lotus. Additionally, the Tianran Hanshi's (天然函昰) cassock and alms bowl Pagoda was built in the Biechuan Temple of Danxia Mountain (丹霞山别传寺) during Qing Kangxi's Reign (see https://mp.weixin.qq.com/s/wnnasdP6zJ5vBhVKL_XRvA, e.g., accessed on 22 May 2023), in addition to the Dangui Jinshi (澹归今释) Pagoda and Zemeng Jinyu (泽萌今遇) Pagoda, which share similar shapes as the Hualin and Liurong Temples (Qiu 2003, pp. 27–31).

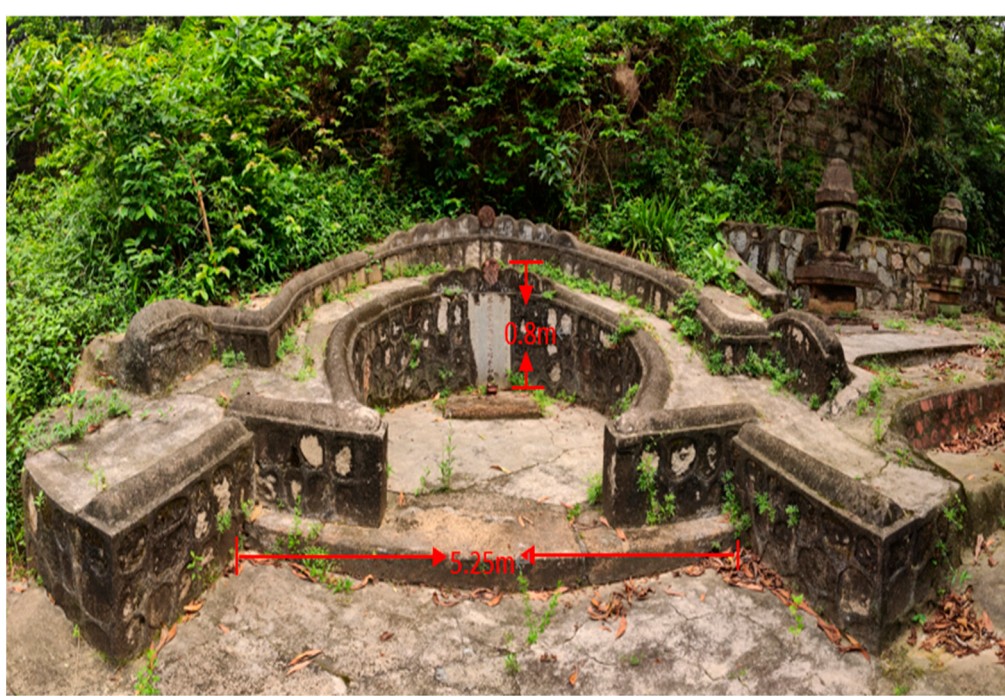

**Figure 6.** Pagoda of Chan Master Ze Anmian of Liurong Temple. Photograph by the author.

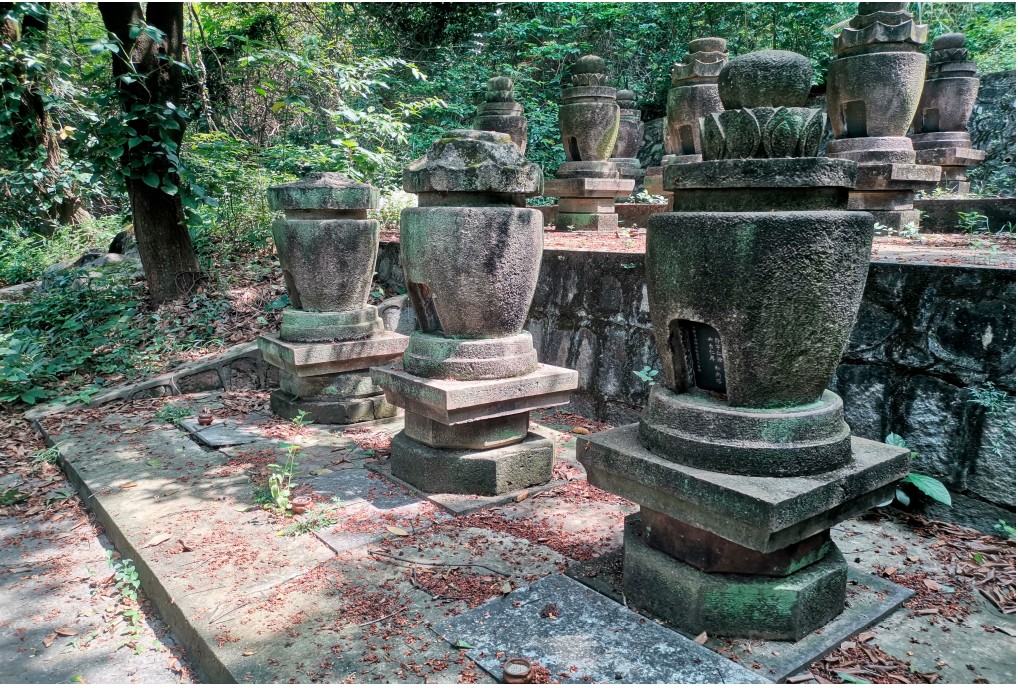

**Figure 7.** Tomb-pagoda Group of Liurong Temple. Photograph by the author.

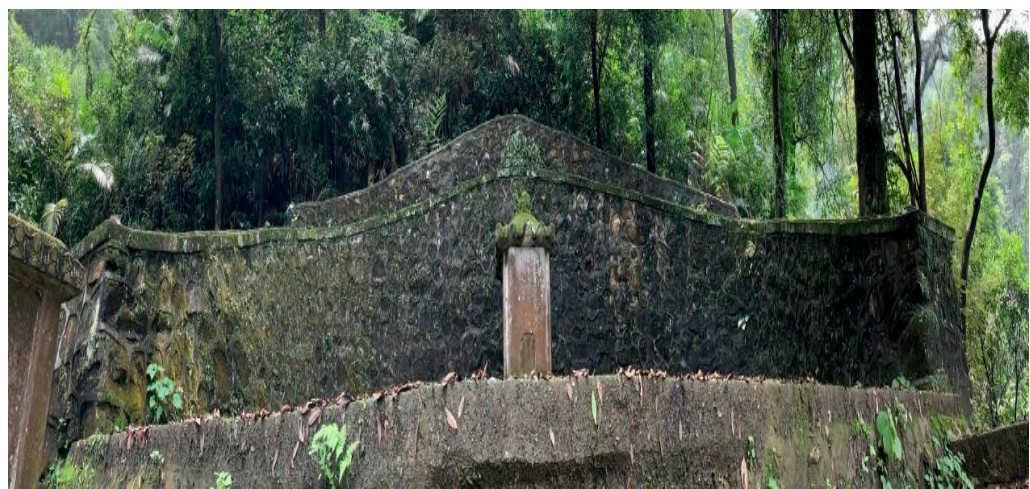

**Figure 8.** Pagoda of the First Abbot of Qingyun Temple in Dinghu Mountain. Photograph by the author.

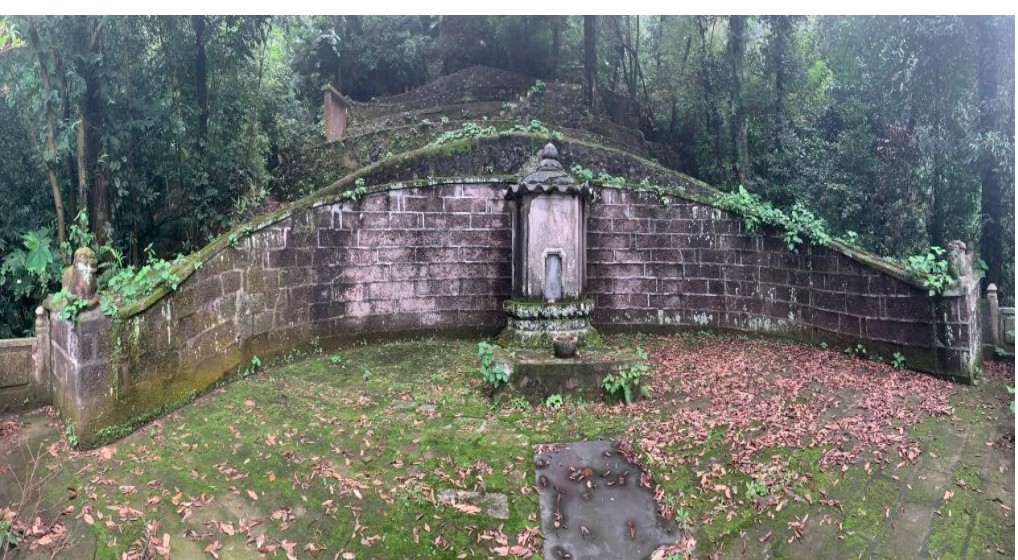

**Figure 9.** Pagoda of the Second Abbot of Qingyun Temple in Dinghu Mountain. Photograph by the author.

　　The Common Pagoda in Haichuang Temple in the Qing Dynasty resembled the ancestral tomb-pagoda in the early Qing Dynasty very much. The former resided in the northwest corner of Haichuang Temple, the corner of Fuchang Garden (福场园) (nowadays the Fuchang Road area of Haizhu District in Guangzhou). There used to be two pagodas (Guo 2016, pp. 547–48). As Constance Frederica Gordon–Cumming pointed out, "One of them, however, may no longer be used, not for lack of room, but because it already contains 4948 sacks of ashes, and Buddhist law forbids the storing of a larger number in one place" (Constance Frederica Gordon-Cumming 1886, p. 90). Unfortunately, both of them were gone. Chinese photographer Li Fang (黎芳) captured a slot in 1883 in the Common Pagoda of Haichuang Temple (see https://rosettaapp.getty.edu/delivery/DeliveryManagerServlet?dps_pid=IE510872, e.g., accessed on 22 May 2023), which showed that it was a single layer tomb-pagoda with a Sumeru pedestal. On top of the base there was square pagoda, and on the front of it there was a stone tablet engraved "Common Pagoda". The top was four corners of the spire, and the surrounding area of the tomb had auxiliary architecture such as loop protection, mountain hands, ridge protecting, etc. The two sides of Guabang had holes that were the entrance for the bone ashes of the dead monks to be thrown into, hence

it was named the Well of Bone Ashes (骨灰井). According to John Henry Gray, "The os-suary, which is a fine piece of masonry, is partially enclosed by a wall, which, in form, greatly resemble the Greek letter Ω" (Gray 2018, p. 80). The Ω referred to the wall, which equaled the ridge protection and the mountain hands. The British Library stores two groups of exporting paintings about Guangzhou, which depicted the scene of the Fuchang Garden of Haichuang Temple in the late 8th century. The shape of the Common Pagoda in the paintings was basically the same as that of the late Qing Dynasty. Obviously, the shape of the Common Pagoda of Haichuang Temple since the Qing Dynasty has not changed much.

The existing Common Pagoda in the Lingnan areas also takes the same form as Haich-uang Temple. For example, in the first year of Yongzheng (1723), the Common Pagoda was built in Lianchi Temple (莲池寺) (Editorial Board 1990, pp. 189–90). Unluckily, the tem-ple was destroyed, and only the pagoda made of stone remains. There were still Sumeru pedestals as the base, bodies in the square, and the treasured stone cover in a lotus shape. The face side of the tablet niche in the front of the pagoda had the following words: "Pu Tong (普同)". Additionally, the Common Pagoda consists of loop protection and ridge protection. The Common Pagodas in the Qing Dynasty exist in many temples, such as the Biechuan Temple in Danxia Mountain (see https://mp.weixin.qq.com/s/ueIOy25Duymp9 sleQ_JYMQ, e.g., accessed on 22 May 2023), the Feilai Temple in Qingyuan (清远飞来寺), and the Kun Iam Temple (普济禅院) in Macao; they are all similar in shape and outlook, so no further elaboration here.[8]

Based on the above, it is shown that the ancestral tomb and Common Pagoda in the Lingnan area in the Qing Dynasty differed distinctly from the traditional tomb-pagoda of Han Buddhism. In the early Qing Dynasty, in Lingnan areas, the master–disciple burial actually originated from the traditional oviform-style pagoda, which was made up of a Sumeru pedestal and the main body. And the difference was its lack of the lotus seat, its body got transformed from an oval to a square, and it also had a lotus-shaped cover added. The traditional oviform-style pagoda belongs to a single whole type. In the early Qing Dynasty, the ancestral tomb-pagoda in Lingnan areas added auxiliary architecture such as loop protection, ridge protection, and mountain hands, which was clearly influenced by the secular Mountain-hands Tomb funeral practices. In Lingnan areas, the Common Pagoda resembled the ancestral tomb-pagoda. This phenomenon existed. As the funeral rites of Han Buddhism gradually degraded with the secular trend, in the late Qing Dynasty, the ancestral tomb-pagoda group in Lingnan areas came closer to the secular fashion. The tomb tablets replaced tomb-pagodas, and hence they were almost the same as the secular Shanshou Tomb, which could be well-improved by the ones in the ancestral tomb-pagoda group of Haichuang Temple, Hualin Temple, and Liurong Temple in the late Qing Dynasty in Baiyun Mountain. These were all typical examples. And, in the late Qing Dynasty in Lingnan areas, on some ancestral tomb tablets there were no such words as "the Pagoda of Chan Master" but "the Tomb of Chan Master" instead. We can find more examples, such as three tomb tablets remaining from the Qing Dynasty in the Lianchi Temple of Guangzhou, the tomb tablet erected during the Qianlong Period, which states "the Pagoda of Great Chan Master Houyun (吼云禅师), the 34th generation of Caodong Sect", and the two tablets erecting during the Daoguang Period called "the Tomb of Old Chan Master" (Li et al. 2008, p. 276).

In the Lingnan area of the Qing Dynasty, why did the Chan monks choose the secular-like Mountain-hands Tomb as their burial rite? According to the study, under rainy and humid climates, the Mountain-hands Tomb has loops and ridges protecting against water washing, hence the tomb could be well-protected (He 1995, p. 72). The Lingnan area is rich in rainfall all year round. The Chan monks' tomb-pagodas have cells underneath, which pose the problem of water sinking in. For example, the Common Pagoda was built in Lvyu Peak (绿玉峰) in its early time when Biechuan Temple in Danxia Mountain was set up, but twenty years later, Chan Master Guyun (古云禅师) wrote that "Due to long years of ele-ments, and the place soaked wet by water, we cannot bear to put bone ashes after cremation

of our dead fellows all these years" (岁久地湿，渐为水所渗入。年来遗蜕虽经荼毗，弗忍投之塔中。) (Chen and Tao 2015, p. 108). So, pagodas were rebuilt in a little mountain beside Jinjiang River (锦江), and we can still see them. In 1873, John Henry Gray visited Haichuang Temple, recording "This ossuary, however, which, on each side, is provided with an aperture, through which the red bags with their contents of human ashes, are thrown, has not, for several years past, been used, owing to the pit or vault, over which it stands, having become full of water" (Gray 2018, pp. 80–81). It is obvious that dropsy has affected the Common Pagodas' normal function. In the surroundings of the Chan monks' tomb-pagodas in the Lingnan area in the Qing Dynasty, people used loop and ridge protection against flood water and had their moon lakes dug into some holes to drain water in order to protect the tomb-pagoda by guarding against possible intense rainfall from damaging the tomb-pagoda and cell. Building a Mountain-hands Tomb needed high levels craftsmanship and was expensive, so it was reserved for the rich to choose from, and ordinary people still piled the earth tomb (He 1995, pp. 70–71). Julia Cox, the wife of John Henry Gray, traveled to Baiyun Mountain and discovered that "Those tombs resembling horseshoes mostly stone houses built by mountain or adobe pressed hard are tombs for wealthy. While the poor people just erected a stone or pile up a hummock" (Julia Cox 2019, p. 17). The abbots of Chan temples in the Lingnan area in the Qing Dynasty were usually abound with enough capital to cover the expenses of building Mountain-hands Tomb-pagodas.

Very interestingly, the process of transforming Chan monks' tomb-pagodas into Shanshou Tombs was also getting underway in some areas in Fujian and Zhejiang. In Shangjinbei Village (上金贝村), Ningde, there is a Yuan Dynasty tomb-pagoda of a Chan Master (see https://www.163.com/dy/article/HJSB3NUF0553A42Q.html, e.g., accessed on 17 June 2023). This freestanding stone pagoda features an inscription on its tower body niche that reads "The Pagoda of Chan Master Cang Haizhu (沧海珠禅师), the third-generation successor of Fori Yuanming who was a Chan master with Golden Kachaya bestowed by the emperor" (御赐金襕佛日圆明第三代沧海珠禅师之塔). Initially, this tomb was mistakenly identified as the burial site of Emperor Jianwen (建文帝) of the Ming Dynasty. However, it is, in fact, the resting place of Chan Master Cang Haizhu, who was the third-generation Dharma heir of the renowned monk Haiyun Yinjian (海云印简) during the early years of the Yuan Dynasty. The pagoda is adorned with a circular tower body resting on a pedestal and follows the distinctive oviform-style design of the Song Dynasty, characterized by its lack of a canopy. Surrounding the pagoda are loop protection, ridge protection, and decorative mountain-shaped motifs. The mountain-shaped motifs feature intricately carved dragon heads, believed to have been added during subsequent restoration efforts in the Qing Dynasty and beyond. The "Common Pagoda for All monks" (十方普同塔) in Wenzhou was completed in Kangxi's 9th year in rein, and a loop, ridge protection, and mountain hands surround the pagoda. (see http://news.66wz.com/system/2016/05/18/104832010.shtml, e.g., accessed on 23 May 2023). During the Ming and Qing Dynasties, in the Wenzhou area, the Shanshou Tombs were in vogue, but the Chan monks' tomb-pagoda style had not been replaced by the Shanshou Tombs in Whenzhou and Fujian, as was the case in Lingnan. Accordingly, different Chan monks' tomb-pagodas in different areas has its own characteristics. Whichever form is taken is affected by many factors commonly working together, such as the natural condition, custom and convention, religion, culture, etc. So, it cannot be generalized.

## 5. Conclusions

Whichever form to take in terms of funeral rites is not only a matter of social conventions, but also the product of the beliefs of ancestors, of souls, and Para. As Buddhism arrived in China, its alien culture of burial influenced China's homeland culture of burial, and they also learned from each other. On one hand, the funeral rituals in Buddhism have gradually become a part of secular funeral rites, manifesting in some special burial forms such as fire burial, exposed-body burial, and companion burial, which were added based

on China's traditional whole-body burial into the earth. There are many Buddhism factors that appeared in tomb decorations and funeral objects (Haibo 2007, pp. 163–99). On the other hand, Han Buddhism also takes secular funeral elements into their burial rituals. Monks follow the secular tradition by wearing mourning garments, using music at funerals, standing as guards at the bier, wailing at funerals, giving a gift to the bereaved family, performing funeral oration, making an epitaph for the dead monk afterward, and even offering a posthumous title to some venerable monks after their departing. *Wu Shan Lian Ruo Sin Shue Bei Yong,* reflecting on the funeral rituals for monks in Five Dynasties, records the "Monk's Five Relations Pictogram" (僧五服图), which corresponds with the "Confucian Five Relations Pictogram" (儒家五服图), exclaiming that disciples observe mourning for their master, and at the funeral, monks wear mourning clothes and wail, express condolences, and perform the offering. In the Song and Ming Dynasties, influenced by the culture of Neo-Confucianism and a set of etiquette, the funeral became closer to the secular one. The funeral rites and the tomb-pagodas were one of the ways to establish a sense of identity. Yirun was criticized over degradation of the secular trend in *Explanation of Monastic Rules of the Monk Baizhang* and emphasized returning to the original Buddhism oriented for funeral rituals. Take the Haichuang Temple of Guangzhou as an example. We find that in the late Qing Dynasty, the over-secularization of funerals in Han Buddhism did not change, and the phenomenon that *Explanation of Monastic Rules of the Monk Baizhang* had criticized still stayed for a long time and was manifested in many forms. And the ancestral tomb-pagoda in the Haichuang Temple of Guangzhou grew more Shanshou-tomb-oriented, with not a slight degree of difference from secular practices.

The lack of Buddhist factors in the funeral rituals of Han Buddhism monks in the late Qing Dynasty shows the fact that Buddhist funeral rituals were eroded by secular cultures, reflecting Buddhism's declining trend in the late Qing Dynasty. Even though different sections of Buddhism were still broadcasting, Buddhist thought lacked creativity in comparison to previous dynasties. The Chan School ushered in a revival scene at the turn of the Ming Dynasty to the Qing Dynasty; when many famous masters came out, temples were built in clusters, and many disciples and believers gathered. However, Chan Thought never innovated. In the late Qing Dynasty, when Han Buddhism further declined, most monks made a living by participating in Buddhist service (经忏佛事, foshih or ching-ch'an). And Buddhism was totally degraded to "the Buddhism for the Death" (死人的佛教), gradually losing its orthodoxy and vigorous tradition. The lives of monks went away, not in the track stipulated by the monastic rules of Chan, but they got closer to secular habitude, and some even became morally degenerate, so the precepts and disciplines almost became a piece of nonsense paper. This article discusses the typical model of the monks' funeral rituals at Haichuang Temple in the late Qing dynasty. In conclusion, the secularization of monks' funeral practices is the epitome of Han Buddhism's extreme decline in the late Qing Dynasty.

**Author Contributions:** Writing—original draft preparation, R.W. and W.C.; writing—review and editing, R.W. and W.C. All authors have read and agreed to the published version of the manuscript.

**Funding:** This research was funded by National Social Science Fund of China (2018), grant number 18CZJ010.

**Data Availability Statement:** Not applicable.

**Conflicts of Interest:** The authors declare no conflict of interest.



**Glossary**

| | |
|---|---|
| Ancestral hall | 祖堂 |
| Ancestral tomb-pagodas | 祖师墓、祖师塔 |
| *An Encyclopedic Dictionary of Buddhism* | 《释氏要览》 |
| An incense burnt cost | 一炷香的时间 |
| Baiyun Mountain | 白云山 |
| Bian Haikuan | 彼岸海宽 (of the Qing Dynasty) |
| Biechuan Temple of Danxia Mountain | 丹霞山别传寺 |
| Boshan | 博山 (another name for Chan Master Wuyi Yuanlai) |
| Burial in a forest | 林葬 |
| Buddhist service | 经忏佛事 (foshih or ching-ch'an) |
| Buddhism for the Death | 死人的佛教 |
| Butterfly Ridge | 蝴蝶岭 |
| Chan Master Cang Haizhu | 沧海珠禅师 (of the Yuan Dynasty) |
| Chan Master Eryan Yingdi | 二严应谛禅师 (of the Qing Dynasty) |
| Chan Maste Guyun | 古云禅师 (of the Qing Dynasty) |
| Chan Master Houyun | 吼云禅师 (of the Qing Dynasty) |
| Chan Master Jinchuan | 今传禅师 (of the Qing Dynasty) |
| Chan Master Jinwu | 今无禅师 (of the Qing Dynasty) |
| Chan Master Wuyi | 无疑禅师 (of the Ming Dynasty) |
| Chan Master Ze Anmian | 则安棉禅师 (of the Qing Dynasty) |
| Changle Temple Village | 长乐寺村 |
| Chiyue | 池月 (of the Ming Dynasty) |
| Confucian Five Relations Pictogram | 儒家五服图 |
| *Chronicle of Haichuang Temple* | 《海幢寺春秋》 |
| Chonggeng Dayi | 崇庚达颐 (of the Qing Dynasty) |
| Common Pagoda | 普同塔 |
| Common Pagoda for All monks | 十方普同塔 |
| *Continuation of The Biographies of Eminent Monks* | 《续高僧传》 |
| Dangui Jinshi | 澹归今释 (of the Qing Dynasty) |
| Dharmakaya | 法身 |
| Drum-shaped stone | 石鼓 |
| Dying Place | 无常院 |
| Eastern Hall | 东堂 |
| Emperor Jianwen | 建文帝 (of the Ming Dynasty) |
| Erlong Valley | 二龙谷 |
| Evaluation and auction of dead monks' relics | 估唱 |
| Four-fundamental streamer | 四本幡 |
| Fuchang Garden | 福场园 |
| Guabang | 挂榜 |
| Guangmou | 光牟 (of the Ming Dynasty) |
| Guest Hall | 客堂 |
| Guike Tomb | 龟壳墓 |
| Haiyun Yinjian | 海云印简 (of the Yuan Dynasty) |
| Heshun Hillock | 和顺岗 |
| Houtu | 后土 |
| Hualin Temple | 华林寺 |
| Hung the portrait to enshrine | 挂真供养 |
| Jiechuang Monastery | 戒幢律寺 |
| Jingu Hillock | 金鼓岗 |
| Jinjiang River | 锦江 |
| Joint burial tomb | 合葬墓 |
| Juhan Yuanhai | 巨涵源海 (of the Qing Dynasty) |
| Kun Iam Temple | 普济禅院 |
| Lianchi Temple | 莲池寺 |
| Liang Peilan | 梁佩兰 (of the Qing Dynasty) |

| | |
|---|---|
| Li Fang | 黎芳 (of the Qing Dynasty) |
| Lihuan Yuanjue | 离幻圆觉 (of the Qing Dynasty) |
| Lingnan | 岭南 (Guangdong Province) |
| Liurong Temple | 六榕寺 |
| Longevity Hall | 延寿堂 |
| Loop protection | 环垄 |
| Lvyu Peak | 绿玉峰 |
| Memorial tablet | 灵位 |
| Monastic Rules | 清规 |
| *Monastic Rules of the Monk Baizhang* | 《敕修百丈清规》 |
| Monk's Five Relations Pictogram | 僧五服图 |
| Monks in charge of temple affairs | 两序执事 |
| Moon lake | 月池 |
| Mountain flanks that are like two hands of a person | 山手 |
| Mountain Hand | 山手 |
| Mountain-shaped motif | 山手 |
| Nirvana Hall | 涅槃堂 |
| Niche | 龛 |
| Offering tea | 奠茶汤 |
| Old Monk Jueshi | 觉师老和尚 (of the Qing Dynasty) |
| Old Monk Kongyin | 空隐老和尚 (of the Qing Dynasty) |
| Oviform-style pagoda | 卵塔 |
| Pagoda courtyard | 塔院 |
| Pagoda of an abbot | 祖师塔 |
| Platform for worshiping | 拜台 |
| Preaching Hall | 法堂 |
| Pu Tong | 普同 |
| Put the whole body in the pagoda | 全身入塔 |
| Qianqiu Temple | 千秋寺 |
| Qihe Daoqiu | 栖壑道丘 (of the Qing Dynasty) |
| Qingyun Chan Temple in Dinghu Mountain Mountain | 鼎湖山庆云寺 |
| *Regulations in Chan Monastery* | 《禅苑清规》 |
| Ridge protection | 护岭 |
| Sai-Kwai-Tong | 西归堂 |
| Shamian | 沙面 |
| Shanshou | 山手 |
| Shangfangshan Monastery | 上方山塔院 |
| Shangjinbei Village | 上金贝村 |
| Shanshou Tomb | 山手墓 |
| Shaolin Temple | 少林寺 |
| Shengchuang | 绳床 |
| Heshun Hillock | 和顺岗 |
| Sickness Hall | 病堂 |
| Spirit tablet | 灵位 |
| *Standby Rules of Chan Forest* | 《禅林备用清规》 |
| Sumeru pedestal | 须弥座 |
| *Summary of Rules of Chan Temples* | 《丛林校定清规总要》 |
| Cemetery of monk | 塔林 |
| Taishi | 太师 |
| *Explanation of Monastic Rules of the Monk Baizhang* | 《百丈清规证义记》 |

| | |
|---|---|
| Tianping Mountain | 天平山 |
| Tianran Hanshi | 天然函昰 (of the Qing Dynasty) |
| Ten-Direction Common Pagoda | 十方普同塔 |
| Three masters in the literature of Lingnan | 岭南三大家 |
| Tomb-pagoda | 墓塔 |
| Tomb of Chair | 椅子坟 |
| Tomb of Folding Chair | 交椅坟 |
| Tomb of the Chair of Taishi | 太师椅墓 |
| Tombstone | 墓碑 |
| Well of Bone Ashes | 骨灰井 |
| Wuben Fayi | 无本法一 (of the Qing Dynasty) |
| *Wu Shan Lian Ruo Sin Shue Bei Yong* | 《五杉练若新学备用》 |
| Xiaocan | 小参 |
| Xigui Hall | 西归堂 |
| Xingxing Hall | 省行堂 |
| Xingzhi Vinayacharya | 性祇律师 (of the Ming Dynasty) |
| Yao Guangxiao | 姚广孝 (of the Ming Dynasty) |
| Yangqu County | 阳曲县 |
| Yirun | 仪润 (of the Qing Dynasty) |
| Yongquan Temple in Gu Mountain | 鼓山涌泉寺 |
| Yunju Yuanyou | 云居元佑 (of the Northern Song Dynasty) |
| Zaisan Hongzan | 在犙弘赞 (of the Ming Dynasty) |
| Zemeng Jinyu | 泽萌今遇 (of the Qing Dynasty) |
| Zengcheng District | 增城区 (of Guangzhou) |
| Zhan Ruoshui | 湛若水 (of the Ming Dynasty) |
| Zhengchijian | 正持鉴 (of the Qing Dynasty) |
| Zhudong Wuwan | 竺东悟万 (of the Ming Dynasty) |

## Notes

[1] For the studies of inhumation and burial in a forest of Chinese Buddhist monks in mediaeval times, refer to Liu (2008a), pp. 183–243. Liu (2008b), pp. 244–90.

[2] For relevant studies on funeral rites of Chan monks in Song and Yuan dynasties, see Cheng (1989), pp. 73–110. Cheng (1995), pp. 121–65. Huang (2018), pp. 213–19. Wang (2013), pp. 278–96.

[3] The original title "*Walks in the City of Canton*" was changed by the translator to "*Seven Days of Guangzhou*".

[4] "'Though the Trikâya be absolutely complete, the limit is not yet found'. 'It is the maturity of the Skandha+ which alone can give perfection'". Given the unavailability of corroborative Chinese historical sources, translating this particular phrase into Chinese poses certain challenges and it just like an eloquent couplet."The yellow strip of paper pasted on to the vertical slide above mentioned bore this inscription: 'The throne of intelligence' of the contemplative philosopher, the Bôdhisatva, the worthy Bikshu 'United Wisdom,' now passed away". Given the unavailability of corroborative Chinese historical sources, translating this particular phrase into Chinese poses certain challenges. So, we translate it under our understanding and the Chinese meaning may be "佛陀座下智者，现已圆寂。".

[5] Gatha is some sentences created by the ritual holder, like poems.

[6] For the shape and structure of Zhan Ruoshui tomb, refer to Hu (2017), pp. 55–66.

[7] The No. 7 Tomb named Yeling Yingxian in the ancestral tomb-pagoda group of Haichuang Temple, established by his disciples in the eighth year of Tongzhi (1869), one of which named "Dahai". The Pagoda of Eryanyingdi should be set up by "the shaved disciple Yuanhai", not "Da Hai". Refer to Xincheng (2008), p. 275.

[8] For the shapes of Common Pagoda of Feilai Temple in Qingyuan and the Kun Iam Temple in Macao, refer to Thomson (2001), p. 36; He (1999), pp. 37–47.

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

### *Secondary Sources*

Chen, Jianhua. 2008. *Collection of Guangzhou Cultural Relics Survey Baiyunshan Volume*. Guangzhou: Guangzhou Publishing House.

Cheng, He Mineo. 1989. Study on Burial of Venerable Monks-Rules of Chan Temples-the Followings of Funeral Procession. In *The Summary of Chan Research Institute of Aichi Gakuin University 17*. Aichi: Aichi Gakuin University.

Cheng, He Mineo. 1995. The Funeral Rites of Chan School. In *Summary of Chan Research Institute of Aizhi Gakuin University 24*. Aichi: Aichi Gakuin University.

Ciyi. 2005. *Fo Guang Dictionary*. Beijing: Beijing Library Press.

Editorial Board. 1990. *Cultural Relics of Guangzhou City*. Guangzhou: Lingnan Arts Press.

Giles, Herbet A. 2007. *From Swatow to Canton*. Edited and Compiled by Bingwei Huang. Shanghai: Fudan University Press.

Gray, John Henry. 2019. *Seven Days in Guangzhou*. Translated by Guoqing Li. Guangzhou: Guangdong People's Publishing House Co.

Haibo. 2007. *The Buddha's Theory of Death: A Study of the Chinese Buddhist View of Death in the Vision of Death Science*. Xian: Shanxi People's Publishing House Co.

He, Bin. 1995. *Funeral Culture of Jiangsu, Zhejiang and Han Nationality*. Beijing: China Minzu University Press.

He, Jianming. 1999. A Brief Discussion on the Buddhist Cultural Relations between Macao and the Mainland in Qing Dynasty: A Case Study on Kun Iam Temple. In *Review of Culture 38*; Macao: Cultural Affairs Bureau of the Macao S.A.R.

Huang, Kui. 2018. *Chinese Chan Rules*. Beijing: Religious and Culture Press.

Huang, Minzhi. 1985. Custom of China's Cremation. In *The Collection of Essays of the Professor of Fu Lecheng: New Theory of Chinese History*. Taibei: Student Book Co.

Hu, Xiaoyu. 2017. The Ming Dynasty Tomb of Zhan Ruoshui tomb, Zengcheng District in Guangzhou. In *The Collection of Archaeology 20*. Beijing: IA CASS.

Julia Cox, Gray. 2019. *A Letter from Guangzhou*. Translated by Xiuying Zou. Guangzhou: Guangdong People's Publishing House Co.

Laracy, Hugh. 2013. Constance Frederica Gordon-Cumming (1837–1924): Traveler, Author, Painter. In *Watriama and Co: Further Pacific Islands Portraits*. Canberra: ANU Press.

Li, Anbao, and Zhengsen Cui. 1999. *Sanjin Ancient Pagoda*. Taiyuan: Shanxi People's Press.

Liu, Shufen. 2008a. Cave Burial in Stone Rooms—Studies on the Burial of Corpse on the Ground in Ancient Buddhism II. In *Ancient Buddhism and Society*. Shanghai: Shanghai Classics Publishing House.

Liu, Shufen. 2008b. Forest Burial—One of the Studies on the Burial of Corpse on the Ground in Ancient Buddhism. In *Ancient Buddhism and Society*. Shanghai: Shanghai Classics Publishing House.

Li, Zhongwei, Zixiong Lin, and Zhimin Cui. 2008. *A Collection of Inscriptions in Guangzhou Temple Temple*. Guangzhou: Guangdong People's Publishing House Co.

Lu, Yiyue. 2018. *Daily Life of Chinese People: Scenery of Rivers and Roads in Fujian*. Translated by Yuejun Zhang, and Weijie Liu. Xiamen: Xiamen University Press.

Qin, Yiran. 2020. The Decline of Chan Buddhism in the Qing Dynasty—A Text Investigation Centered on the Funeral Activities of the Abbot of the Qing Dynasty. In *Tian Fu Xin Lun 2*. Chengdu: Sichuan Provincial Federation of Social Sciences.

Qiu, Jiang. 2003. Pagoda for Monks of Biechuan Temple in Danxia Mountain in the Qing Dynasty. In *Lingnan Literature and History 2*. Guangzhou: Research Institute of Culture and History, the People's Government of Guangdong Province.

Thomson. 2001. *Old China on the Camera: Journey of John Thomson's Travels*. Translated by Boren Yang, and Xianping Chen. Beijing: China Photography Press.

Wang, Dawei. 2013. *Study on Chan Rules of Song and Yuan Dynasties*. Beijing: Religious and Culture Press.

Wang, Ronghuang. 2018. On the Power, Status and Secularization of the Jungle Residence in the Ming and Qing Dynasties. In *Studies on Religion 1*. Chengdu: Institute of Taoism and Religious Culture, Sichuan University.

Wang, Xuebin. 2020. Common Pagoda: Research on the Monk Burial System under the Simple Ecological Concept. In *Journal of Huanghe S&T University 4*. Zhengzhou: Huanghe S & T University.

Xincheng. 2008. *Chronicle of Haichuang Temple (《海幢寺春秋》)*. Guangzhou: Huacheng Publishing House.

Zhang, Yuhuan. 2000. *China Tower*. Taiyuan: Shanxi People's Press.

Zhou, Xing. 1995. Chair Tomb and Turtle shell Tomb. In *Funeral Culture of Jiangsu and Zhejiang Han Nationality*. Beijing: China Minzu University Press.

Zhou, Yukai. 2013. The Old Monk has died into a new pagoda—Discussion on the system of Buddhist pagoda and its funeral culture. In *Studies on Religion 3*. Chengdu: Institute of Taoism and Religious Culture, Sichuan University.

