# Peer review of "A Study on the Funerals of the Han Buddhist Monks of Lingnan during the Late Qing Dynasty via the Haichuang Temple in Guangzhou"

_religions, doi:10.3390/rel14070924_

Round 1
Reviewer 1 Report
The paper, A Study on the Funeral of Han Buddhist Monks in Late Qing Dynasty though Haichuang Temple in Guangzhou, selected the funeral problems of monks in the late Qing Dynasty which is rarely studied, therefore it has an innovative topic. By taking account of the westerners’ historical records on the ceremonies and manners of monks’ funerals in Haichuang Temple in Late Qing Dynasty, and by comparing those records with writings in The Regulations in Zen Monastery, Monastic Rules of the Monk Baizhang, especially the Monastic Rules of the Monk Baizhang in Qing Dynasty and other related records of Monastic Rules of Zen , the paper argues that influenced by the secular funeral culture in Lingnan Area, the tombs of the monks in Zen Temples represented by the Haichuang Temple had already shown a trend of transition to Shanshou tombs in the early Qing Dynasty; and in the late Qing Dynasty, traditional forms of pagodas were completely abandoned in some Lingnan Zen Temples, and the patriarchs’ tomb-pagodas were almost the same as secular Shanshou Tombs. The paper also points out that the degradation of the funeral culture of Han Buddhism in the late Qing Dynasty reflects the declining trend of Buddhism. In the paper, the author adopts first-hand materials such as people's insights and compares them with Monastic Rules of Zen, its research method is a scientific one and convincing conclusion is made.
However, the research of this paper only focuses on Haichuang Temple, some other Buddhist Temple in Lingnan Area or Buddhist temples in Guangdong, Fujian and Zhejiang provinces, and still lacks research on ceremonies and manners of monks’ funeral in other parts of China at that time. In addition, the paper says that according to westerner’s records, there was a narrow strip of yellow paper pasted on either side of a monk’s niche and some sentences were wrote on: ' Though the Trikâya be absolutely complete, the limit is not yet found.' 'It is the maturity of the Skandha+ which alone can give perfection.' Can you find the original text? Besides, there is some doubt to the sentence that during the funeral of a monk of Haichuang Temple, the abbot was coming with his string of 108 beads in one hand, and he stopped in front of the altar and coffin, and there prostrated himself thrice, each time knocking his head upon the ground thrice.
The paper says that according to westerner’s records, there was a narrow strip of yellow paper pasted on either side of a monk’s niche and some sentences were wrote on: ' Though the Trikâya be absolutely complete, the limit is not yet found.' 'It is the maturity of the Skandha+ which alone can give perfection.' Can you find the original text?
Author Response
Point 1: However, the research of this paper only focuses on Haichuang Temple, some other Buddhist Temple in Lingnan Area or Buddhist temples in Guangdong, Fujian and Zhejiang provinces, and still lacks research on ceremonies and manners of monks’ funeral in other parts of China at that time.
Response 1: The title of the thesis has been revised to "Research on Burial Practices of Han Buddhist Monks of Lingnan in Late Qing Dynasty," and the fourth section of the main text has been supplemented with discussions on various other forms of ancestral pagodas during the Ming and Qing Dynasties.
Point 2: In addition, the paper says that according to westerner’s records, there was a narrow strip of yellow paper pasted on either side of a monk’s niche and some sentences were wrote on: ' Though the Trikâya be absolutely complete, the limit is not yet found.' 'It is the maturity of the Skandha+ which alone can give perfection.' Can you find the original text?
Response 2: Given the unavailability of corroborative Chinese historical sources, translating this particular phrase into Chinese poses certain challenges, as elucidated in the provided annotation, illustrating it as an eloquent couplet.
Point 3: There is some doubt to the sentence that during the funeral of a monk of Haichuang Temple, the abbot was coming with his string of 108 beads in one hand, and he stopped in front of the altar and coffin, and there prostrated himself thrice, each time knocking his head upon the ground thrice.
Response 3: Citing English literature as the primary reference, since there is a dearth of Chinese historical documentation for comparative analysis, preserving the original English rendering is deemed more judicious in this context.

Reviewer 2 Report
This research paper "A Study on the Funeral of Han Buddhist Monks in Late Qing Dynasty through Haichuang Temple in Guangzhou" is well written. However, there are several awkward sentences and miss-spellings to be corrected.
1) line3: misspelling in the title: "A Study on the Funeral of Han Buddhist Monks in Late Qing Dynasty through Haichuang Temple in Guangzhou"
2) In the Introduction, using Chinese characters in parenthesis for English pronunciation is better. Also, it is suggestive to make a glossary for the important Chinese terms at the end of research paper by following the guideline of the Journal Religions.
3) line 77-80: I wonder whether the "relocation of the sick and critically ill" would be considered as "the funeral activities for ordinary monks"? It is necessary to clarify ‘the sick’ and ‘critically ill.’
4) line 131-133: “The yellow strip of paper pasted on to the vertical slide above mentioned bore this inscription: 'The throne of intelligence’ of the contemplative philosopher, the Bôdhisatva, the worthy Bikshu ‘United Wisdom,’ now passed away.”14 => This quotation does not make sense. It is necessary to provide the written Chinese.
5) line 134: Could you provide some information for Gatha? What kind of Gatha is recited?
6) line 154: The sentence “After that, they began to burial site” is better to be clarified.
7) Which statement is correct between A and B?
A : line 212-213: “the niche was kept three to seven days”(停棺 三七日)
B: Line 261-262: The Monastic Rules of the Monk Baizhang in Qing Dynasty stipulated that when the abbot was dead, “his niche will be placed there for 21 days”;
8) line 244-245: What is the meaning “offering sacrifices and soup and tea(奠茶汤),…” ? In Buddhism, ‘offering sacrifices’ is prohibited. It is necessary to explain it.
9) line 317-318: Elaborate about the different sizes in threes, fives, sevens, and nines : “the temple courtyard and the pagoda courtyard had different sizes in threes, fives, sevens, and nines.”
10) In Chapter 4, Figures of pagodas are well elaborated. However, the photos are small in sizes, it is difficult to clearly figure out the differences.
11) In the conclusion, the author generalized his conclusion in lines 657-667 without providing detailed written sources.
12) There are several other missed spellings and repeated words to be corrected: For examples: p. pp. see see, etc.
Author Response
Point 1: Line3: misspelling in the title: "A Study on the Funeral of Han Buddhist Monks in Late Qing Dynasty through Haichuang Temple in Guangzhou"
Response 1: The title of the thesis has been revised to "Research on Burial Practices of Han Buddhist Monks of Lingnan in Late Qing Dynasty via Haichuang Temple in Guangzhou."
Point 2: In the Introduction, using Chinese characters in parenthesis for English pronunciation is better. Also, it is suggestive to make a glossary for the important Chinese terms at the end of research paper by following the guideline of the Journal Religions.
Response 2: The glossary has been made in the revised manuscript.
Point 3: Line 77-80: I wonder whether the "relocation of the sick and critically ill" would be considered as "the funeral activities for ordinary monks"? It is necessary to clarify ‘the sick’ and ‘critically ill.’
Response 3: "Relocation of the sick and critically ill" has been revised to ”move the critically ill monks to the Xigui Hall.” Only monks who are critically ill will move to Xigui Tang.
Point 4: Line 131-133: “The yellow strip of paper pasted on to the vertical slide above mentioned bore this inscription: 'The throne of intelligence’ of the contemplative philosopher, the Bôdhisatva, the worthy Bikshu ‘United Wisdom,’ now passed away.”14 => This quotation does not make sense. It is necessary to provide the written Chinese.
Response 4: Given the unavailability of corroborative Chinese historical sources, translating this particular phrase into Chinese poses certain challenges.So,we translate it under our understanding and the Chinese meaning may be”佛陀座下智者,现已圆寂。” The sentence has been elucidated in the provided annotation.
Point 5: Could you provide some information for Gatha? What kind of Gatha is recited?
Response 5: The gatha is some sentences created by the ritual holder, like poems and also has been elucidated in the provided annotation.
Point 6: The sentence “After that, they began to burial site” is better to be clarified.
Response 6: It has been revised to “After that, they began to go to the cremation furnace.”
Point 7: Which statement is correct between A and B?
A : line 212-213: “the niche was kept three to seven days”(停棺 三七日)
B: Line 261-262: The Monastic Rules of the Monk Baizhang in Qing Dynasty stipulated that when the abbot was dead, “his niche will be placed there for 21 days”;
Response 7: It is B. All the statement has been revised.
Point 8: What is the meaning “offering sacrifices and soup and tea(奠茶汤),…” ? In Buddhism, ‘offering sacrifices’ is prohibited. It is necessary to explain it.
Response 8: That means offering tea in front of the niche and‘offering sacrifices’has been revised.
Point 9: Elaborate about the different sizes in threes, fives, sevens, and nines : “the temple courtyard and the pagoda courtyard had different sizes in threes, fives, sevens, and nines.”
Response 9: It has been revised to” The temple courtyard had different systems, there can be threes pagodas, fives pagodas, sevens pagodas, or nines pagodas in the pagodas courtyard.”
Point 10: In Chapter 4, Figures of pagodas are well elaborated. However, the photos are small in sizes, it is difficult to clearly figure out the differences.
Response 10: All the pictures in the article have been enlarged, and some have been marked with size, which is easier to figure out the differences.
Point 11: In the conclusion, the author generalized his conclusion in lines 657-667 without providing detailed written sources.
Response 11: Written sources “Wei Mian. Summary of Rules of Chan Temples, see CBETA 2023.Q1, X63, no. 1249.”“Zheng Xian. Standby Rules of Chan Forest, see CBETA 2023.Q1, X63, no. 1250. ”“Zheng Xian. Standby Rules of Chan Forest, see CBETA 2023.Q1, X63, no. 1250.” CBETA is the abbreviation of Chinese Buddhist Electronic Text Association, X for Shinsan Zakuzokyo, the number 63 for volume, which also has been elucidated in the provided annotation.
Point 12: There are several other missed spellings and repeated words to be corrected: For examples: p. pp. see see, etc.
Response 12: Thank you for correction, we have conducted multiple inspections and have corrected the wrong spelling and made a highlight in the text.
